# Changes in Bone Mineral Density and Trabecular Bone Score over Time between Vegetarian and Non-Vegetarian Middle-Aged and Older Women: A Three-Year Retrospective Medical Record Review

**DOI:** 10.3390/ijerph19042445

**Published:** 2022-02-20

**Authors:** Tzyy-Ling Chuang, Malcolm Koo, Mei-Hua Chuang, Chun-Hung Lin, Chin-Huan Huang, Yuh-Feng Wang

**Affiliations:** 1Department of Nuclear Medicine, Dalin Tzu Chi Hospital, Buddhist Tzu Chi Medical Foundation, Chiayi 622401, Taiwan; b8601139@tmu.edu.tw; 2School of Medicine, Tzu Chi University, Hualien 970374, Taiwan; cmh618@ms32.hinet.net (M.-H.C.); linch51.leo@gmail.com (C.-H.L.); 3Graduate Institute of Long-Term Care, Tzu Chi University of Science and Technology, Hualien 970302, Taiwan; m.koo@utoronto.ca; 4Faculty of Pharmacy, National Yang Ming Chiao Tung University, Taipei 112304, Taiwan; 5MacKay Junior College of Medicine, Nursing, and Management, New Taipei 25245, Taiwan; 6Department of General Surgery, Dalin Tzu Chi Hospital, Buddhist Tzu Chi Medical Foundation, Chiayi 622401, Taiwan; 7Department of Nutrition Therapy, Dalin Tzu Chi Hospital, Buddhist Tzu Chi Medical Foundation, Chiayi 622401, Taiwan; dalinrd@tzuchi.com.tw

**Keywords:** vegetarian, dietary habit, bone mineral density, trabecular bone score, women

## Abstract

The effect of a vegetarian diet on bone health remains controversial. This retrospective medical record review compared changes in bone mineral density (BMD) and trabecular bone score (TBS) between vegetarian and non-vegetarian middle-aged and older women who underwent two general health examinations (T1 and T2) that were approximately three years apart. Generalized estimating equations were used to compare the change in lumbar spine and bilateral hip BMD and TBS over time. At T1, the mean age of the patients was 56.6 years (standard deviation 9.7 years) and the mean interval between T1 and T2 was 2.7 years. For women aged 40–55 years, compared with non-vegetarians, vegetarians were significantly associated with a larger reduction in lumbar spine BMD (*p* < 0.001) and left hip femoral neck BMD (*p* = 0.015) over the three-year interval. On the contrary, changes in BMD were not significant at any site in women aged ≥ 56 years. Moreover, the changes in BMD and TBS over the three-year interval did not significantly differ between vegetarian and non-vegetarian women aged 65–90 years. In conclusion, for women aged 40–55 years, vegetarian diets reduced bone quantity, as measured by BMD, but not bone quality, as measured by TBS.

## 1. Introduction

Osteoporosis is a systemic skeletal disease characterized by low bone density (BMD) and microarchitectural deterioration of bone tissues, resulting in significant morbidity and mortality [1,2]. Osteoporosis is three-times more common in women than in men, because women have a lower peak bone mass and due to hormonal changes that occur during menopause [3]. Postmenopausal women are predominantly disposed to suffering from common bone fractures, such as vertebral and hip fractures [4].

Although the rate of bone loss varies according to the anatomic site, cortical and trabecular bone loss progresses with aging in both sexes. In women, there is an inverse non-linear association between age and all BMD measurements [5]. Although there are few changes in BMD during premenopause or early perimenopause, it can decrease substantially in late perimenopause, with an average loss of 0.018 and 0.010 g/cm^2^ per year in the spine and hip, respectively. In the postmenopausal period, the rates of loss in the spine and hip increase to 0.022 and 0.013 g/cm^2^ per year, respectively [6]. Perimenopause is defined as the transition period immediately prior to menopause [7]. This period starts, on average, four years before the last menstrual period, at any age from 42 to 52 years [8].

Both the macrostructure and microarchitecture of bone contribute to bone strength. Thinned trabecula and diminished connectivity, observed in the bones of postmenopausal women, can result in a reduction in the load-bearing capacity of older bones [9]. With age, women could lose 35–50% of trabecular bone mass and 25–30% of cortical bone mass [10]. The most commonly used and reliable technique for measuring BMD is dual-energy X-ray absorptiometry (DXA) of the hip and spine. Based on different attenuation characteristics of bone and soft tissue exposed to X-ray radiation at two peak energies, BMD measurements expressed that grams of mineral, primarily calcium, per square centimeter (g/cm^2^) of the scanned bone can be obtained from mathematical algorithms.

However, BMD is an assessment of the quantity but not of the quality of bone, and it does not provide information about the trabecular structure of the bone. A recent development in the measurement of trabecular microarchitecture is the trabecular bone score (TBS), introduced in 2008 by Pothuaud et al. [11]. TBS is an indirect evaluation of three-dimensional bone microarchitecture based on the use of experimental variograms of two-dimensional projection images obtained during a DXA scan. Previous studies have shown that TBS is correlated with trabecular number, trabecular thickness, connectivity density, and structure model index [12,13].

TBS has been explored to assess the risk of osteoporotic fracture, independent of BMD [14,15]. A meta-analysis of 17,809 participants in 14 prospective population-based cohorts reported that TBS was significantly associated with major osteoporotic fractures in postmenopausal women. In addition, TBS remained an independent and significant predictor of fracture risk when adjusted for Fracture Risk Assessment Tool (FRAX) 10-year probability [16]. A bibliometric analysis of TBS publications, indexed in the Web of Science database from 2008 to 2019, identified 430 original and review articles. The number of articles increased steadily from 2008 to 2019, reaching 80 articles in 2019 alone [17]. Furthermore, a review of TBS use in clinical practice concluded that TBS has an additional role apart from BMD in assessing osteoporotic fracture risk in postmenopausal women and in men over 50 years of age [18].

Previous research suggested that vegetarians had lower BMD than non-vegetarians [19]. The prospective EPIC-Oxford cohort study with approximately 18 years of follow-up showed that, compared to meat eaters, vegans had higher risks of total, hip, leg, and vertebral fractures, while fish eaters and vegetarians had higher risks of hip fractures [20]. In addition, a recent meta-analysis of 17 cross-sectional studies that included 13,888 patients revealed that both vegetarians and vegans exhibited lower lumbar spine, femoral neck, and whole-body BMDs than omnivores [21]. However, other studies showed that both BMD and risk of osteoporotic fractures were similar between vegetarians and omnivores [22,23]. Given the conflicting literature on the effect of vegetarian diet on BMD [24] and the lack of studies of its effect on TBS, this study aimed to longitudinally compare changes in BMD and TBS at the spine and hip in vegetarian and non-vegetarian middle-aged and older women.

## 2. Materials and Methods

### 2.1. Study Participants and Study Variables

The study protocol was approved by the Institutional Review Board of the study hospital (IRB No. B11001010), which waived the requirement to obtain informed consent from patients.

In this retrospective review of medical records, women aged 40 to 90 years who had undergone a general health examination from June 2014 to July 2020 at Dalin Tzu Chi Hospital, Taiwan, were reviewed. Those who had undergone a health examination twice within an approximate three-year interval were identified and included in the analysis. The two time points (first and second health examination) are referred to as T1 and T2, respectively, in the following description.

Electronic health records of patients at T1 were examined to obtain their age, body mass index (BMI), systolic blood pressure [SBP], diastolic blood pressure [DBP], and blood test data. Blood samples were analyzed on a Beckman Coulter DxC 700 AU Clinical Chemistry System (Mishima, Japan) or Beckman Coulter Automated Chemistry Analyzer AU5800 (non-sterile) (Mishima, Japan) and Sysmex automatic XN hematology analyzer (Kobe, Japan) for the following measurements: total cholesterol (TCH), triglycerides (TG), high-density lipoprotein cholesterol (HDL-C), low-density lipoprotein cholesterol (LDL-C), fasting glucose, albumin (ALB), alkaline phosphatase (ALP), and estimated glomerular filtration rate (eGFR).

### 2.2. Vegetarian and Non-Vegetarian Status

Vegetarian status was defined on the basis of a question on dietary habits ascertained during each health examination. The assessment of vegetarian status is a routine procedure in all health examinations conducted in our hospital, and the status is verbally confirmed by trained staff. Vegans, lacto-vegetarians, and ovo-lacto-vegetarians were all considered vegetarians.

### 2.3. Measurements of BMD and BMD T-Score

Absolute BMD values were obtained for the lumbar spine and bilateral hips (total and femoral neck regions using DXA on a DiscoveryWi DXA system (Hologic Inc., Marlborough, MA, USA). Patients whose BMD-measured areas containing metal materials were excluded. Extreme values (>4 standard deviations [SD] from the norm) in the differences in BMD between T1 and T2 were also excluded. Five BMD outcome variables were analyzed: lumbar spine, right hip femoral neck, left hip femoral neck, right hip total, and left hip total.

### 2.4. Measurement of Trabecular Bone Score

The TBS values of patients at T1 and T2 were retrospectively computed using the iNsight software version 3.0.2.0 (MedImaps, Geneva, Switzerland) of the spine DXA files from the patient database. The TBS was quantified from local variations in pixel intensity and derived from experimental variograms obtained from the gray levels of a DXA image.

### 2.5. Statistical Analysis

The patients were divided into three subgroups according to age at their second health examination (40–55 years, 56–64 years, and 65–90 years) to represent the premenopausal, perimenopausal, and postmenopausal periods of this population [18]. Categorical variables were summarized as a frequency with percentage. Continuous variables were summarized as means with SD. Paired sample t-test or McNemar’s test was used, as appropriate, to compare the difference in variables between T1 and T2 for each of the three age groups. In addition, generalized estimating equations (GEE) procedure was used to evaluate changes in BMD and TBS between T1 and T2. First, separate GEE models for age, BMI, and each of the laboratory measurements were conducted to identify variables at T1 that were significantly associated with TBS and the five BMD outcome variables. Significant variables were then included in a final GEE model with vegetarian status, time (T1 and T2), and an interaction term between the two. A significant interaction term between vegetarian status and time (vegetarian × time) would indicate differential rates of change in the outcome variable between vegetarian and non-vegetarian groups.

To evaluate the effects of misclassification of vegetarian status, we performed a simulation-based sensitivity analysis. We randomly selected 10% and 20% of the vegetarians in our data and reassigned them as non-vegetarians. This process was repeated 1000 times for the 10% and 20% misclassification levels to generate 2000 simulated datasets. The original GEE model was re-estimated using the simulated datasets. The reason for the choice of the direction of misclassification was based on previous research suggesting that people might self-identify as vegetarian but occasionally eat meat. A survey study of 243 vegetarians reported that more than half of them had violated their vegetarian eating behaviors, mainly to avoid disrupting existing or expected social dynamics around omnivores [25].

All statistical analyses were performed using PASW Statistics for Windows, Version 18.0 (SPSS Inc., Chicago, IL, USA). A two-sided *p*-value < 0.05 was considered as statistically significant.

## 3. Results

The mean age of the female patients was 56.6 (SD 9.7) years at T1 and 59.3 (SD 9.7) years at T2. The mean interval between T1 and T2 was 2.7 years (SD 0.3, range 2.1–3.6). Overall, 60% of the patients were vegetarian. Table 1 shows the demographic and clinical characteristics of patients stratified into age groups. There were no significant differences between groups in the proportion of vegetarians. BMI increased significantly from T1 to T2 in all three age groups. Regarding bone measurement, TBS and BMD consistently decreased in all three age groups at T2 compared with T1 at all sites, except for right hip total BMD. There were also significant changes in ALP and eGFR in all three age groups, and in DBP, HDL-C, LDL-C, fasting blood glucose, albumin, and total cholesterol in some age groups.

Results from the GEE assessing the association between the changes in BMD and TBS over time between vegetarian and non-vegetarian women in the three age groups are shown in Table 2, Table 3 and Table 4. These tables show a significant interaction term between being a vegetarian and time, meaning that the change in the outcome variables over time (T1 to T2) was significantly different between vegetarians and non-vegetarians. For example, the interaction term was significant (*p* < 0.001) in women aged 40–55 years in the lumbar spine and in the left hip femoral neck BMD (*p* = 0.015) (Table 2).

Figure 1 and Figure 2 show the scatter plots of the lumbar spine and left femoral neck BMD, respectively, of vegetarian and non-vegetarian women in the three age groups. Only the plots for the lumbar spine and left femoral neck BMD were constructed because the interactions between time and diet in other BMD sites and TBS were not statistically significant. In women 40–55 years, a vegetarian diet was significantly associated with a greater decrease over time in both lumbar spine BMD (*p* < 0.001) and left hip femoral neck BMD (*p* = 0.015) (Figure 2). However, for those 56–64 years and 65–90 years, a vegetarian diet was not significantly associated with changes in any of the BMD sites over time. In addition, a vegetarian diet was not significantly associated with changes in TBS over time in any of the three age groups.

Table 5 showed the results of simulation-based sensitivity analysis of 10% and 20% misclassification in the vegetarian status on the change in lumbar spine BMD, left hip femoral neck BMD, and trabecular bone score over time. The lumbar spine BMD and left hip femoral neck BMD were chosen out of the five BMD variables because these were the only two with significant vegetarian × time interaction in the 40–55 years age group. Results of the simulation showed that with 10% or 20% of women misclassified themselves as vegetarians, the *p* values of the vegetarian × time interaction term obtained from the GEE models were similar between those from the original data and the simulated datasets. The only exception was the *p* value of the left hip femoral neck BMD, which increased from 0.032 in the original data to a mean value of 0.061 calculated from 1000 simulated datasets with 20% of misclassification in the vegetarian status.

## 4. Discussion

In this retrospective study based on longitudinal health examination data, we found that, in women during the perimenopausal period (aged 40–55 years), the adoption of a vegetarian diet was associated with a significantly faster loss of bone mass compared with a non-vegetarian diet. However, no significant decrease in bone quality, as defined by microarchitectural deterioration of bone tissue based on TBS, was observed. These findings remained robust in simulation-based sensitivity analysis.

In Taiwan, the mean age of onset of natural menopause is reported as 50.2 years (SD 4.0) [26]. Therefore, our youngest age group (40–55 years) should include women in their perimenopausal period. The prevalence of osteoporosis at the lumbar spine and femoral neck in Taiwanese women increases from 8.3% and 5.2% in those aged 40–49 years to 16.1% and 24.0% in those aged 80 years and older, respectively [27]. Bone mass reaches its peak in women between the ages of 20 and 30 years. Then, BMD decreases gradually and then continues to decline rapidly after menopause [28]. The transition to menopause represents a limited window of opportunity in time to intervene in the rapid loss of bone and microarchitectural damage found in later years [29].

Moreover, in our youngest age group, the interaction between vegetarian diet and time was significant in the BMD of the left but not the right hip femoral neck. The exact reason as to why the left side (assumed non-dominant side) decreased at a significant faster rate in those aged 40–55 years is not clear. It is still controversial whether leg dominance affects BMD of the hip regions [30]. In clinical practice, the measurement of the non-dominant hip is preferred based on the assumption that the non-dominant side is less physically active and, therefore, exposed to less stress and impact, resulting in lower BMD.

The estimated annual rate of premenopausal BMD loss in US women is 0.7–1.3% at the lumbar spine [31,32] and 0.3% at the femoral neck [33]. In perimenopausal women, it is > 2% in the lumbar spine [34] and 0.6% in the femoral neck [35]. In Taiwanese women, Shaw et al. calculated annual BMD loss in the lumbar spine as 0.2% in those aged 30–33 years and 0.6% in those aged 40–49 years after 5–6 years of follow-up [36]. In our study, the annual rate of BMD loss at the lumbar spine was 1.3% in women aged 40–55 years, 0.8% in those 56–64 years and 0.6% in those 65–90 years. The annual rate of loss of BMD in the femoral neck was 1.4% in women aged 40–55 years and 1.2% in women aged 56–90 years.

The exact mechanism behind the timing of menopause-associated bone loss has not been conclusively elucidated. It was thought to be associated with estrogen deficiency. However, a recent prospective study suggested that the increased rate of anovulatory cycle, in the presence of adequate level of estrogen, might be a causal factor for perimenopausal bone loss [37]. Compared to the present study, it is possible that a decrease in ovulatory rate did not affect bone quality, as reflected by TBS. Additional studies will be required to clarify the implications of ovulatory decline and progesterone deficiency in perimenopausal bone loss.

The trabecular bone is a porous type of bone tissue found at the epiphyses and metaphyses of long bones as well as in vertebral bodies [38]. Unlike cortical bone, which serves as support, trabecular bone functions to shift mechanical load. The vertebral body is the main trabecular bone site, and vertebral compression fractures are a key characteristic of osteoporosis [39]. Women lose about 50% of their trabecular bone and 30% of their cortical bone during their lifetime, about half during the first 10 years after menopause [40]. BMD is rapidly lost with the beginning of menopause due to the loss of ovarian function and decreased estrogen production. In our study, a significantly faster decrease in BMD was observed only among vegetarians aged 40–55 years, but not in the two older age groups. However, no significant changes in TBS were found in any of the three groups, suggesting that bone microarchitecture and, therefore, bone mechanical resistance was not adversely affected by a vegetarian diet. A possible reason could be that the trabecular bone is more active in the bone remodeling process than the cortical bone and, consequently, less mineralized [38].

A sedentary lifestyle and certain dietary habits can impact bone health and the risk of osteoporosis. A recent meta-analysis of 20 studies with 37,134 participants showed that vegetarian and vegan diets were associated with lower BMD in the femoral neck and lumbar spine compared to an omnivore diet. In addition, vegans were more prone to fractures than omnivores [41]. An earlier meta-analysis of nine studies with 2749 individuals revealed that BMD was approximately 4% lower in vegetarians than in omnivores at both the femoral neck and the lumbar spine. The authors concluded that this magnitude of the change in BMD was clinically insignificant [19]. Moreover, compared with non-vegetarian postmenopausal women, ovo-lacto-vegetarians of the same age showed no differences in cortical and trabecular BMD [42]. Furthermore, a study of 1600 women in Southwestern Michigan reported that those who had followed an ovo-lacto-vegetarian diet for at least 20 years had only 18% less bone mineral by age 80 compared to paired omnivores who had 35% less bone mineral [43]. A cross-sectional study of 1865 adult male and female Taiwanese patients found no significant differences in BMD of vegetarian and non-vegetarian men or women [44]. Nevertheless, a study of 258 postmenopausal Taiwanese women who were engaged in long-term vegan vegetarianism observed them to have a higher risk of exceeding the lumbar spine fracture threshold and classified them as having osteopenia of the femoral neck [45]. Similarly, a community-based cross-sectional study of vegetarian women aged 70–89 years showed that the BMD in the spine was similar for vegetarians and omnivores, but the BMD in the hip was significantly lower in vegetarians [46]. The different results of various studies may be partly attributed to different degrees of strictness in adhering to a vegetarian diet (vegan, lacto-vegetarian, or ovolacto-vegetarian) between study populations and the different substitutes for meat products consumed. Nevertheless, vegetarian foods generally have lower levels of saturated fats and cholesterol and higher amounts of dietary fiber and phytochemicals, all of which can promote health in general [47] as well as bone health [48].

The main strength of the present study was the first to report the effect of vegetarian diet on the change in TBS over time. In addition, this study had a large sample size with relatively large proportion of vegetarians. The reason for the high number of vegetarians was that many of those who underwent general health examinations were volunteers from the Buddhist Tzu Chi Foundation, and Buddhists are encouraged to consume a vegetarian diet. Nevertheless, our study also had limitations due to the use of medical records. First, we selected individuals who had undergone two rounds of health examination in an approximately three-year interval. In other words, we did not follow those who had undergone only the first round of a health examination, which might result in selection bias. Second, the types of vegetarian diet and the duration of vegetarian diets could not be ascertained from the records and, hence, could not be analyzed or adjusted for in a statistical model. More detailed dietary information was not available due to the constraints of the length of the standard questionnaire used in the general health examination. As the possibility of misclassification of vegetarian status is impossible to eliminate, we conducted a simulation-based sensitivity analysis to assess the impact of misclassified vegetarian status on our findings. Results of the simulation showed that with 10% or 20% of the women in our study misrepresenting themselves as vegetarian, our conclusion remains valid. No significant decrease in bone quality was observed, as defined by TBS, in women in any of the three age groups. Third, pharmacological treatment used by the patients was not available. However, we included a number of blood biochemical parameters in the statistical evaluation as potential confounders, which should minimize the effects of comorbidity on our results.

## 5. Conclusions

In this retrospective medical record review study, we found that a vegetarian diet could reduce bone quantity, as reflected by BMD, but not bone quality, as reflected by TBS in perimenopausal Taiwanese women over a three-year time interval. On the contrary, a vegetarian diet did not affect the change in either bone quantity or bone quality in women aged 56 years or older. Future studies should explore the effects of different types and durations of vegetarian diet on bone health.

## Figures and Tables

**Figure 1 ijerph-19-02445-f001:**
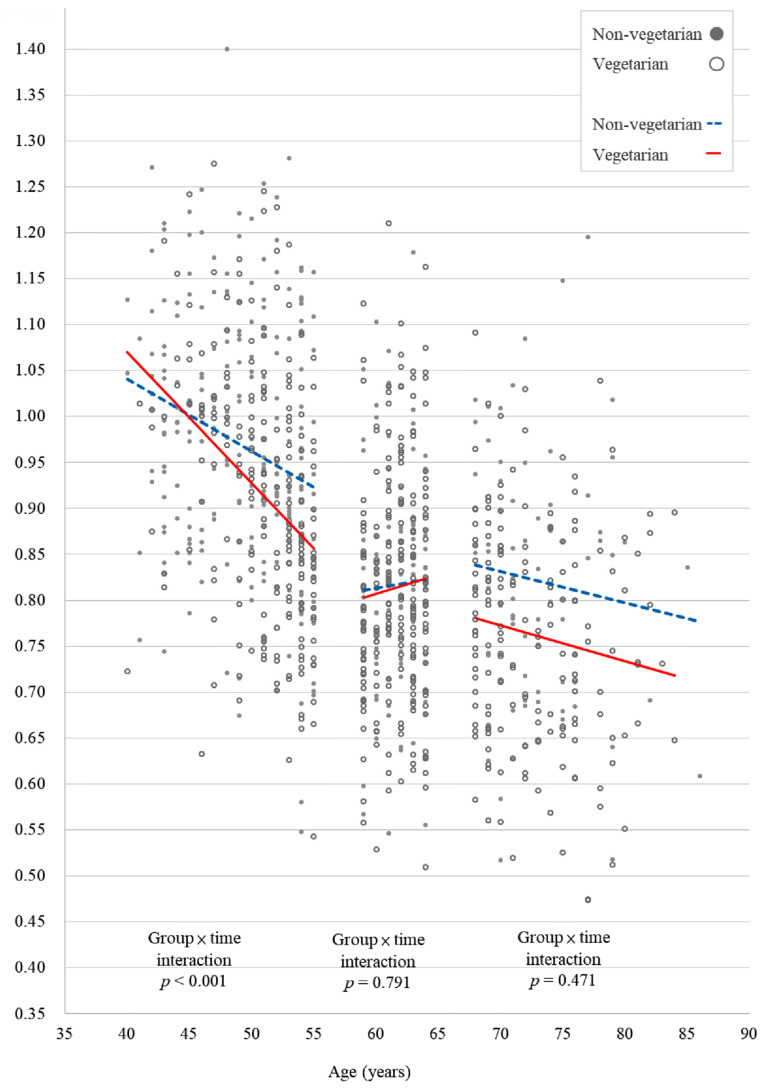
A scatter plot of the effect of the interaction between time and diet on lumbar spine bone mineral density in the three age groups of vegetarian and non-vegetarian women.

**Figure 2 ijerph-19-02445-f002:**
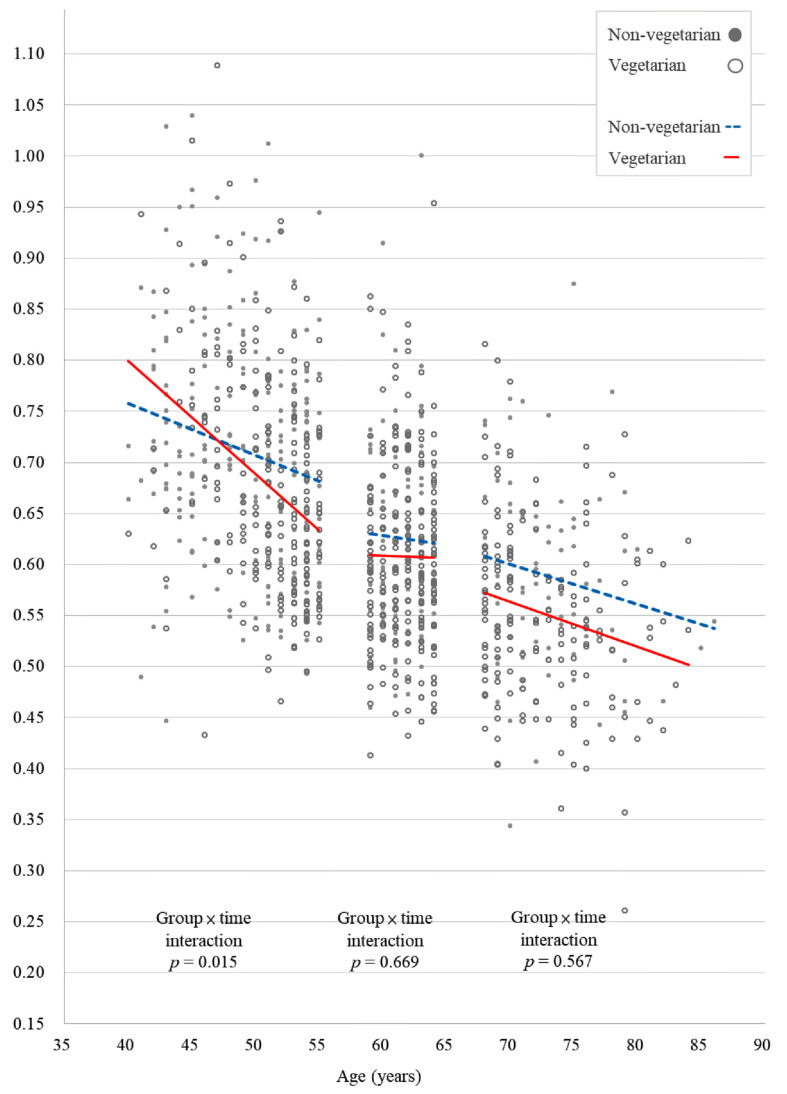
A scatter plot of the effect of the interaction between time and diet on left femoral neck bone mineral density in the three age groups of vegetarian and non-vegetarian women.

**Table 1 ijerph-19-02445-t001:** The demographic and clinical characteristics of the patients, stratified into three age groups.

Variable				Age Group								
	All (*n* = 1049)			40–55 Years (*n* = 458)			56–64 Years (*n* = 344)			65–90 Years (*n* = 247)		
	T1	T2	*p*	T1	T2	*p*	T1	T2	*p*	T1	T2	*p*
Vegetarian, *n* (%)	629 (60.0)	629 (60.0)	>0.999	224 (48.9)	221 (48.3)	0.788	237 (68.9)	237 (68.9)	>0.999	168 (68.0)	171 (69.2)	0.749
Age (years)	56.2 (9.7)	59.2 (9.7)	-	47.0 (3.9)	50.0 (3.9)	-	58.6 (1.7)	61.6 (1.7)	-	69.8 (4.1)	72.8 (4.1)	-
BMI (kg/m^2^)	23.5 (3.3)	23.9 (3.5)	<0.001 *	23.3 (3.6)	23.9 (3.7)	<0.001 *	23.5 (3.1)	23.9 (3.2)	<0.001 *	23.8 (3.2)	24.2 (3.4)	<0.001 *
SBP (mmHg)	126.5 (20.7)	126.1 (19.8)	0.459	120.3 (19.5)	119.9 (18.3)	0.572	128.0 (19.8)	127.5 (18.7)	0.623	135.9 (20.3)	135.6 (20.0)	0.841
DBP (mmHg)	71.9 (11.0)	71.4 (10.8)	0.049 *	71.7 (11.5)	71.4 (11.5)	0.511	72.0 (10.7)	71.6 (10.6)	0.400	72.4 (10.4)	71.0 (9.4)	0.032 *
HDL-C (mg/dL)	53.3 (14.6)	52.7 (14.6)	0.016 *	55.8 (15.4)	55.2 (15.4)	0.130	50.9 (13.0)	50.0 (13.2)	0.037 *	52.0 (14.4)	51.7 (14.3)	0.570
LDL-C (mg/dL)	118.2 (31.5)	114.8 (30.6)	<0.001 *	112.7 (30.0)	113.3 (29.5)	0.541	124.8 (32.2)	118.9 (32.0)	<0.001 *	119.4 (31.6)	111.7 (30.2)	<0.001 *
Fasting blood glucose (mg/dL)	105.0 (21.7)	102.9 (21.7)	<0.001 *	100.4 (17.5)	99.0 (19.3)	0.003 *	107.3 (25.0)	105.1 (22.5)	0.026 *	110.3 (22.1)	107.1 (23.8)	0.006 *
Albumin (g/dL)	4.3 (0.3)	4.4 (0.2)	<0.001 *	4.3 (0.3)	4.4 (0.2)	0.003 *	4.3 (0.3)	4.4 (0.2)	0.091	4.3 (0.3)	4.3 (0.2)	0.064
ALP (IU/L)	79.4 (24.8)	70.6 (20.3)	<0.001 *	70.2 (21.8)	66.3 (19.1)	<0.001 *	87.7 (25.2)	74.4 (20.6)	<0.001 *	85.0 (23.7)	73.0 (20.5)	<0.001 *
eGFR (mL/min/1.73 m^2^)	117.5 (22.4)	99.9 (22.8)	<0.001 *	124.5 (20.8)	105.1 (21.2)	<0.001 *	115.8 (20.9)	101.3 (23.6)	<0.001 *	106.8 (23.0)	88.5 (20.4)	<0.001 *
TCH (mg/dL)	188.8 (35.8)	186.6 (35.6)	0.011 *	183.8 (33.7)	185.9 (34.1)	0.107	194.6 (37.1)	189.9 (37.9)	0.001 *	189.9 (36.5)	183.3 (34.7)	0.001 *
Triglycerides (mg/dL)	104.6 (56.3)	104.3 (61.7)	0.828	93.3 (57.6)	94.5 (66.2)	0.558	114.6 (56.7)	111.6 (59.6)	0.245	111.4 (49.2)	112.3 (52.7)	0.760
Lumbar spine BMD (g/cm^2^)	0.889 (0.150)	0.863 (0.150)	<0.001 *	0.982 (0.130)	0.943 (0.141)	<0.001 *	0.834 (0.118)	0.815 (0.122)	<0.001 *	0.795 (0.125)	0.780 (0.128)	<0.001 *
Right hip femoral neck BMD (g/cm^2^)	0.654 (0.114)	0.626 (0.113)	<0.001 *	0.715 (0.109)	0.681 (0.110)	<0.001 *	0.628 (0.092)	0.603 (0.091)	<0.001 *	0.579 (0.091)	0.554 (0.092)	<0.001 *
Right hip total BMD (g/cm^2^)	0.779 (0.124)	0.795 (0.120)	<0.001 *	0.831 (0.119)	0.841 (0.119)	0.001 *	0.761 (0.108)	0.778 (0.104)	<0.001 *	0.707 (0.112)	0.733 (0.108)	<0.001 *
Left hip neck femoral BMD (g/cm^2^)	0.659 (0.115)	0.637 (0.114)	<0.001 *	0.721 (0.109)	0.696 (0.111)	<0.001 *	0.634 (0.091)	0.614 (0.089)	<0.001 *	0.581 (0.091)	0.563 (0.091)	<0.001 *
Left hip total BMD (g/cm^2^)	0.777 (0.123)	0.756 (0.122)	<0.001 *	0.829 (0.117)	0.806 (0.118)	<0.001 *	0.759 (0.106)	0.737 (0.105)	<0.001 *	0.705 (0.111)	0.690 (0.110)	<0.001 *
TBS	1.342 (0.104)	1.311 (0.107)	<0.001 *	1.415 (0.084)	1.379 (0.093)	<0.001 *	1.304 (0.077)	1.275 (0.083)	<0.001 *	1.260 (0.077)	1.235 (0.082)	<0.001 *

ALP: alkaline phosphatase; BMD: bone mineral density; BMI: body mass index; DBP: diastolic blood pressure; eGFR: estimated glomerular filtration rate; HDL-C: high-density lipoprotein cholesterol; LDL-C: low-density lipoprotein cholesterol; SBP: systolic blood pressure; T1: First general health examination; T2: Second general health examination; TCH: total cholesterol; TBS: trabecular bone score. All values are mean and standard deviation unless stated otherwise. * *p* < 0.05.

**Table 2 ijerph-19-02445-t002:** Results of generalized estimating equations for the effects of a vegetarian dietary habit on the change in BMD and trabecular bone score over time in female patients aged 40–55 years (*n* = 458).

Variable	Lumbar Spine BMD		Right Hip Femoral Neck BMD		Right Hip Total BMD		Left Hip Femoral Neck BMD		Left Hip Total BMD		Trabecular Bone Score	
	β	*p*	β	*p*	β	*p*	β	*p*	β	*p*	β	*p*
Vegetarian × time	−0.018	<0.001 *	−0.007	0.063	−0.009	0.133	−0.009	0.015 *	−0.010	0.099	−0.002	0.705
Vegetarian	−0.005	0.489	−0.003	0.520	−0.008	0.248	−0.010	0.051	−0.004	0.629	0.003	0.611
Time	−0.015	0.004 *	−0.018	<0.001 *	0.015	0.006 *	−0.008	0.071	−0.017	0.005 *	−0.017	<0.001 *
Age	−0.008	<0.001 *	−0.006	<0.001 *	−0.003	0.009 *	−0.006	<0.001 *	−0.005	<0.001 *	−0.007	<0.001 *
BMI	0.010	<0.001 *	0.008	<0.001 *	0.012	<0.001 *	0.008	<0.001 *	0.013	<0.001 *	-	-
HDL-C	-	-	-	-	-	-	-	-	0.005	0.034 *	-	-
Glucose	-	-	-	-	0.004	0.029 *	0.004	0.034 *	-	-	-	-
ALB	-	-	−0.018	<0.001 *	-	-	−0.017	<0.001 *	−0.024	0.002 *	-	-
ALP	−0.008	<0.001 *	−0.003	0.002 *	−0.005	<0.001 *	−0.003	0.001 *	-	-	−0.006	<0.001 *
eGFR	-	-	-	-	-	-	-	-	−0.004	0.004 *	-	-
TG	0.001	0.012 *	-	-	0.001	0.037 *	-	-	0.002	0.003 *	-	-

ALB: albumin (g/dL); ALP: alkaline phosphatase (per 10 IU/L); β: regression parameter coefficient estimate; eGFR: estimated glomerular filtration rate (per 10 mL/min/1.73 m^2^); Glucose: fasting blood glucose (per 10 mg/dL); HDL-C: high-density lipoprotein cholesterol (per 10 mg/dL); TG: triglycerides (per 10 mg/dL). * *p* < 0.05.

**Table 3 ijerph-19-02445-t003:** Results of generalized estimating equations for the effects of a vegetarian dietary habit on the change in BMD and trabecular bone score over time in female patients aged 56–64 years (*n* = 344).

Variable	Lumbar Spine BMD		Right Hip Femoral Neck BMD		Right Hip Total BMD		Left Hip Femoral Neck BMD		Left Hip Total BMD		Trabecular Bone Score	
	β	*p*	β	*p*	β	*p*	β	*p*	β	*p*	β	*p*
Vegetarian × time	−0.002	0.791	0.001	0.766	−0.003	0.724	−0.002	0.669	0.010	0.169	0.003	0.743
Vegetarian	−0.004	0.524	0.003	0.496	0.004	0.648	−0.002	0.670	−0.007	0.380	−0.002	0.815
Time	−0.030	<0.001 *	−0.033	<0.001 *	0.011	0.077	−0.023	<0.001 *	−0.040	<0.001 *	−0.037	<0.001 *
BMI	0.009	<0.001 *	0.007	<0.001 *	0.012	<0.001 *	0.008	<0.001 *	0.011	<0.001 *	-	-
SBP	-	-	-	-	-	-	-	-	-	-	−0.004	0.005 *
ALB	-	-	−0.019	0.002 *	-	-	-	-	-	-	-	-
ALP	−0.007	<0.001 *	−0.003	0.001 *	−0.003	0.036 *	-	-	-	-	−0.005	<0.001 *
eGFR	-	-	-	-	-	-	-	-	−0.001	<0.001 *	-	-
TCH	-	-	-	-	-	-	−0.001	0.002 *	-	-	-	-

ALB: albumin (g/dL); ALP: alkaline phosphatase (10 IU/L); β: regression parameter coefficient estimate; eGFR: estimated glomerular filtration rate (mL/min/1.73 m^2^); SBP: systolic blood pressure (per 10 mmHg); TCH: total cholesterol (per 10 mg/dL). * *p* < 0.05.

**Table 4 ijerph-19-02445-t004:** Results of generalized estimating equations for the effects of a vegetarian dietary habit on the change in BMD and trabecular bone score over time in female patients aged 65–90 years (*n* = 247).

Variable	Lumbar Spine BMD		Right Hip Femoral Neck BMD		Right Hip Total BMD		Left Hip Femoral Neck BMD		Left Hip Total BMD		Trabecular Bone Score	
	β	*p*	β	*p*	β	*p*	β	*p*	β	*p*	β	*p*
Vegetarian × time	0.006	0.471	−0.001	0.786	−0.004	0.668	0.003	0.567	−0.010	0.306	0.007	0.465
Vegetarian	−0.025	0.005 *	−0.008	0.132	−0.005	0.610	−0.010	0.085	0.002	0.860	−0.013	0.154
Time	−0.021	0.002 *	−0.014	0.008 *	0.037	<0.001 *	−0.011	0.046 *	−0.00002	0.998	−0.023	0.007 *
Age	-	-	−0.004	0.007 *	−0.004	0.011 *	−0.004	0.004 *	−0.004	0.014 *	−0.002	0.031 *
BMI	0.010	<0.001 *	0.006	<0.001 *	0.013	<0.001 *	0.006	<0.001 *	0.012	<0.001 *	-	-
SBP	-	-	−0.003	0.010 *	-	-	-	-	-	-	-	-
ALB	-	-	−0.017	0.012 *	-	-	-	-	−0.022	0.047 *	-	-

ALB: albumin (g/dL); SBP: systolic blood pressure (per 10 mmHg). β: regression parameter coefficient estimate; * *p* < 0.05.

**Table 5 ijerph-19-02445-t005:** Simulation-based sensitivity analysis of 10% and 20% misclassification in the vegetarian status on the change in lumbar spine BMD, left hip femoral neck BMD, and trabecular bone score over time.

Age Group (Years)	Mean of *p* Value of the Vegetarian × Time interaction term (95% Confidence Interval)								
	Original Data			10% Misclassification inVegetarian Status			20% Misclassification in Vegetarian Status		
	Lumbar Spine BMD	Left Hip Femoral Neck BMD	Trabecular Bone Score	Lumbar Spine BMD	Left Hip Femoral Neck BMD	Trabecular Bone Score	Lumbar Spine BMD	Left Hip Femoral Neck BMD	Trabecular Bone Score
40–55	<0.001 *	0.015 *	0.705	0.001 (0.001, 0.002)	0.032 (0.030, 0.034)	0.699 (0.687, 0.710)	0.005 (0.004, 0.006)	0.061 (0.056, 0.066)	0.667 (0.654, 0.681)
56–64	0.791	0.669	0.743	0.711 (0.699, 0.723)	0.331 (0.319, 0.343)	0.726 (0.714, 0.738)	0.650 (0.635, 0.664)	0.386 (0.370, 0.402)	0.667 (0.652, 0.681)
65–90	0.471	0.567	0.465	0.589 (0.575, 0.604)	0.693 (0.679, 0.706)	0.542 (0.527, 0.557)	0.590 (0.573, 0.606)	0.637 (0.622, 0.652)	0.544 (0.528, 0.561)

* *p* < 0.05.

## Data Availability

The data used to support the findings of this study are available from the corresponding author upon request.

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
