# Peer review of "Changes in Bone Mineral Density and Trabecular Bone Score over Time between Vegetarian and Non-Vegetarian Middle-Aged and Older Women: A Three-Year Retrospective Medical Record Review"

_ijerph, 2022, doi:10.3390/ijerph19042445_

Round 1

Reviewer 1 Report

This is an interesting study on the relationship between changes in the periods before and after menopause and dietary types.

Although it was metioned in the Discussion section, the definition of vegetarian is not clear, so it is difficult to determine the correlation of dietary and BMD. In addition, I'd recommend the study subjects need to be limited to Taiwanese women in conclusion.

In Table 1, appropriate labeling for statistically significant values/variables of the table may help the reader to understand easily. It is confirmed that there is a significant difference in some variables, such as fasting blood glucose, LDL, HDL, etc, so it'd be good to have sufficient discussion. 

Reviewer 2 Report

I highly rate the article, but I did not find any data regarding the inclusion in the analysis of comorbidities (1) and the pharmacological treatment used by patients (2). The study should be supplemented with all available parameters for both of these issues.
In addition, the following minor problems should be mentioned:
In the abstract, information about the standard deviation of the age of the group should be included (3).
In table 1, data for the whole group should also be added (4).
I may also have a question about the credibility of the declaration of vegetarianism in the respondents. How it was tried to verify and increase it. (5)

Reviewer 3 Report

This manuscript repots a retrospective medical record review, investigating the difference between vegetarian and non-vegetarian middle-aged and older women on the changes in their bone mineral density (BMD) and trabecular bone score (TBS). The changes in BMD and TBS were followed for 2 years. The samples are divided into three subgroups; 40‒55 years (458 volunteers), 56‒64 years (344 volunteers), and 65‒90 years (247 volunteers). This manuscript is potential for publication in IJERPH after a few minor corrections.

  1. The Introduction should be improved to guide wide readers on this specific topic.
  2. Line 54-55; This sentence should be “In recent 54 years, the trabecular bone score (TBS) has been increasingly used …”
  3. Table 1. “Trabecular bone score” should be replaced by “Trabecular bone score (TBS)” to clarify the abbreviation in the text.
  4. Line 134-135: “TBS and BMD consistently decreased …” Is there any proposed reason for the exception for higher right hip total BMD in all three age groups?
  5. Table 2-4. Authors should define β and p, and discuss the significant different p values in Figure 1-2.
  6. Figure 1-2. Presentation of the figures can be improved, so that they can be easily seen.
  7. Line 205-. Authors should discuss and compare the results with those on perimenopausal bone loss which is recently published in Diagnostics 2022, 12, 305.
  8. https://doi.org/10.3390/diagnostics12020305.

Round 2

Reviewer 2 Report

II have no further comments. The authors made corrections according to my suggestions.